# Influence of Sintering Parameters on Spectroscopic Properties of BMW: Eu^3+^ Ceramic Materials Prepared by HPLT Technique

**DOI:** 10.3390/ma15217410

**Published:** 2022-10-22

**Authors:** Natalia Miniajluk-Gaweł, Robert Tomala, Bartosz Bondzior, Przemysław Jacek Dereń

**Affiliations:** Institute of Low Temperature and Structure Research, Polish Academy of Science, Okólna 2, 50-422 Wroclaw, Poland

**Keywords:** double perovskite, Eu^3+^ ions, ceramic materials, HPLT technique, WLEDs

## Abstract

In this work, Ba_2_MgWO_6_: Eu^3+^ (BMW: Eu^3+^) ceramic materials with a double perovskite structure were sintered using the High-Pressure Low-Temperature sintering (HPLT) technique. As part of the research, the influence of pressure (CP), sintering temperature (CT), and sintering time (CTS) on the structure and luminescence of the doped BMW were determined. Structural analysis via XRD and SEM + EDS and spectroscopic analysis via emission and excitation spectra, decay time, and absorption spectra of the obtained ceramics were performed. Dense double perovskite ceramics were obtained with a cubic structure with optimal sintering parameters: T = 500 °C, *p* = 8 GPa, and t = 1 min. The increase in temperature caused an increased extinction of the luminescence due to the diffusion of carbon into the ceramics. The increase in pressure led to the formation of the amorphous phase, which increased the speed of non-radiative transitions and also led to the extinction of the luminescence. The increase in sintering time from 1 to 3 min enhanced the luminescence output, but when the ceramic was sintered for 5 min, the luminescence was quenched, most likely by increasing the rate of the non-radiative process, as evidenced by reduced decay time.

## 1. Introduction

Ba_2_MgWO_6_ (BMW) double perovskite is a good candidate for making transparent ceramics because it possesses desirable properties, such as cubic structure with the space group *Fm−3m*, the theoretical density of δ_th_—7.2 g/cm^3^, and most importantly, excellent compaction ability which may be related to the low friction and rearrangement forces encountered during compaction [1,2]. Ceramic materials have aroused great interest in recent years; therefore, many research centers are looking for ideal materials in order to obtain transparent ceramics. Physicochemical properties of ceramics significantly exceed many other materials in terms of resistance to rapid temperature changes and the ability to work in difficult conditions (corrosive or chemical environment, high temperature, or high pressure). Importantly, the light scattering phenomenon also occurs in ceramics, but it is much smaller than in the currently used phosphors [1,2].

However, the literature is very poor in describing Ba_2_MgWO_6_ ceramic materials with a double perovskite structure [1,2,3,4,5,6]. There are only a few works that mainly focus on describing their structural properties [3,4,5,6] and merely two [1,2] about the luminescent properties of these ceramic materials. In the work [3], a Ba_2_MgWO_6_ ceramic was obtained by the traditional solid-state sintering to study its internal properties to elucidate the microscopic mechanism of its dielectric response. In another work [4], phase formation, microstructure, and microwave dielectric properties of Ba_2_(Mg_1−2x_Y_2x_W_1−x_Ti_x_)O_6_ ceramics with an ordered perovskite structure were investigated. L.A.R. De Aguiar et al. [5] studied the structural, microstructural, and mechanical properties of Ba_2_MgWO_6_ ceramics for communication technology. Chen et al. [6] investigated how sintering temperature affects the microstructures and microwave dielectric properties of Ba_2_MgWO_6_ ceramics. Other works [1,2] concern the preparation of BMW double perovskites using the new sintering method Spark Plasma Sintering (SPS). In the work [1], for the first time, it was possible to obtain translucent BMW and BMW: Ce^3+^ ceramics by the SPS method. The SPS method seems to be an excellent alternative to HP (hot pressing) or HIP (hot isostatic pressing) to obtain a completely dense and fine-grained transparent ceramic at only 1350 °C for 5 min under uniaxial pressure of 50 MPa. The resulting ceramic has a regular structure and is characterized by high density, which would enable it to absorb radiation with excellent efficiency, and it could therefore be used as a medical imaging scintillator. The paper [2] presents studies of the spectroscopic properties of ceramics BMW:1% Nd^3+^ and BMW:1% Eu^3+^ under the influence of X-ray radiation. The results of both BMW ceramics showed that their spectra were in the visible region after X-ray excitation, offering a novel approach to the material as a scintillator.

Optimization of obtaining transparent ceramics is the subject of research all over the world due to their interesting applications. The greatest technological difficulty in this process is to ensure the high level of compaction required to achieve perfect clarity. The key to obtaining the necessary almost complete densification is the appropriate microstructure of the samples in the green state. Accordingly, more attention is now paid to pre-firing treatment. The main features of the highly sinterable powder compact are the small and narrow pore distribution dimensions, and this configuration must be strictly maintained throughout the macroscopic sample volume. Therefore, the steps of powder preparation, processing, and forming must be optimized and correlated to ensure proper management of the grinding of powder aggregates and uniform compact distribution of particles resulting from breaking aggregates [7]. Therefore, in order to meet the requirements, this paper presents the results of optimization of the sintering conditions by selecting the appropriate pressure (1–8 GPa), temperature (500–1350 °C), and sintering time (1–5 min), while maintaining a constant rate of temperature increase.

However, on the other hand, the high transparency of the material causes low optical scattering and lowers the absorption cross section [8]. BMW, due to its thermal properties [9] and the distances between dopant atoms, which allow doping with more active ions without concentration quenching, is a good candidate for use as a matrix for light sources pumped by semiconductor diodes, such as White LEDs.

In this work, BMW: Eu^3+^ powders were obtained by the solid-state method. The fabrication method of BMW:Eu^3+^ ceramics with the use of the High-Pressure Low-Temperature (HPLT) sintering technique was described earlier by Fedyk et al. [10]. Thus far, this method has been used mainly for the preparation of the following ceramics: YAG:Nd^3+^ [10,11], Y_2_O_3_:Er^3+^, Yb^3+^ [12], La_0_._9_A_0_._1_MnO_3_ (A: Li, Na, K) [13], YAG:Yb^3+^ [14,15], and YAG:Eu^3+^ [16,17]. The HPLT technique is very similar to the commonly used SPS sintering technique, except that the advantage of the HPLT technique is that the current flow is continuous, thanks to which it is possible to increase the pressure up to a maximum of 8 GPa. The SPS method is generated by a high-intensity impulse current in the conductive matrix, and the sample is heated by transferring energy through graphite, hence it is not possible to obtain high pressure. The HPLT method allows for a quick and cheap way to obtain a material with high density and high mechanical strength, while maintaining the spectroscopic properties of the precursor. In this work, the HPLT method for sintering double perovskites with Ba_2_MgWO_6_ structure was used for the first time. Structural and spectroscopic analysis of the obtained ceramics was performed. Dense double perovskite ceramics with a cubic structure were obtained.

## 2. Experimental

### 2.1. Sample Preparation

The subject of our research is ceramic materials with a double perovskite structure, doped with Eu^3+^: Ba_2_Mg_0_._985_Eu_0_._01_Li_0_._005_WO_6_ (BMW:Eu). Polycrystalline powders were prepared by the conventional high-temperature solid-state reaction method. A detailed description of individual stages of synthesis can be found in the recently published work [2]. Additionally, Figure 1 shows a solid-state synthesis scheme for obtaining polycrystalline BMW: Eu powders.

In our previous work [18], we described in detail why Eu^3+^ ions occupy the site of Mg^2+^ in the BMW host and not at the Ba^2+^ positions. Rather, it is not possible for the ions of the trivalent dopant to occupy the W^6+^ position due to the too large charge difference. First, taking into account the ion radius values (Table 1), the smallest differences are between the impurity ions and Mg^2+^ ions. Secondly, cell volumes for all series of Eu^3+^ concentrations were calculated by indexing and compared with the parameters of the undoped host. The unit cell volume increases with increasing Eu^3+^ doping concentration. We can therefore infer that the larger ion is replacing the host cation. Additionally, emission spectra were measured at 10 K. The obtained results showed that the electric dipole-type transitions are very weak compared to the dominant magnetic dipole-type ^5^D_0_ → ^7^F_1_, which indicates a high symmetry of the site, confirming the position of Eu^3+^ ions in the highest O_h_ symmetry of the Mg site. Substitution of a divalent ion with a third oxidation state lanthanide ion can cause defects in the matrix, which occur to compensate the created charges locally in the host. For this reason, charge compensation was applied through the addition of lithium. The calculations were performed by substituting lithium in the position of magnesium, the same as the dopant ion.

The obtained powder was then sintered using the High-Pressure Low-Temperature (HPLT) technique [10]. In Figure 2 a schematic representation of the high-pressure cell is presented. The pellet is located in the center of the cell and surrounded by hexagonal boron nitride (BN), which serves as an insulator. The graphite tube is a resistance heater and a ceramic made of CaCO_3_ is situated between two anvils and plays the role of pressure container. The force excreted by the anvils creates a quasi-isostatic pressure in the range of 1–8 GPa. Before sintering, the green was formed from the powder by cold pressing in a cylindrical cell under the pressure of 25 MPa of diameter 5 mm. The sintering conditions were optimized by selecting the appropriate pressure (1–8 GPa), temperature (500–1350 ℃), and sintering time (1–5 min), while maintaining a constant temperature increase rate (see Table 2). The last stage of the process was machining and polishing.

In this work, materials synthesized by the solid-state method Ba_2_Mg_0_._99_Eu_0_._01_WO_6_ (BMW:Eu) with changing parameters of pressure, temperature, and time of sintering, are designated as BMW:Eu -CP, BMW:Eu-CT, and BMW:Eu-CTS, respectively.

### 2.2. Research Techniques

X-ray diffraction (XRD) patterns were recorded with X’Pert ProPANalytical X-ray diffractometer (Malvern Panalytical, Malvern, UK), working in the reflection geometry, using CuKα radiation (λ = 1.54056 Ǻ). The data were collected in a 2Θ range from 10° to 90° with a step of 0.026°.

A scanning electron microscope (FEI NOVA NanoSEM 230, Brno, Czech Republic, equipped with EDAX Genesis XM4 detector, Mahwah, NJ, USA) was used to characterize the morphology and chemical composition of the samples. The SEM images, as well as the EDS spectra and maps, were recorded with an accelerating voltage of 5 kV and 20 kV, respectively.

Emission and excitation spectra at 300 K were recorded using an Edinburgh FLS 980 spectrofluorometer system (Livingston, UK) with a light excitation produced by a 450 W xenon lamp (Edinburgh Photonics, Livingston, UK) and detected by an R928P photomultiplier (Oxford Instruments, Abingdon, UK).

Decay times were collected using a 150 W pulse Xenon lamp as the excitation source.

For the measurements of the reflectance absorption spectra, the Varian Cary 5E UV–vis-NIR spectrophotometer was used (Agilent, Santa Clara, CA, USA).

## 3. Results and Discussion

Figure 3 shows X-ray diffraction patterns of BMW:Eu ceramic materials, which differ in sintering parameters. All ceramic materials have a double perovskite structure of Ba_2_MgWO_6_. The diffraction peaks of the samples match well with the ICDD card no. 70-2023. Each of the materials has a cubic double perovskite structure, with the space group *Fm−3m* and the following crystallographic structural parameters: lattice parameter a = 8.1120 Å, unit cell volume V = 533.81 Å^3^, and Z = 4. Figure 4 shows the structure of a representative double perovskite Ba2MgWO6 material. This material adopts the cubic structure with a rock salt lattice of corner-shared octahedra of MgO_6_ and WO_6_, with Ba cations positioned in the cubic 12-fold coordination sites, as shown in Figure 4 (MgO_6_—blue polyhedra and WO_6_—dark red polyhedra) [18].

The parameters of the sintering process did not change the structure of the obtained materials but affect their phase character. All the ceramics were practically single-phase. Only in the case of ceramic materials for which the sintering time and temperature were changed did single reflections from the additional phase of BaWO_4_ (ICDD card no. 43-0646, Figure 3) appear. The wide band in the range of 13–23 Θ, with the maximum intensity at 16.8 Θ, does not indicate the presence of an amorphous structure but is only a result of technical apparatus problems, i.e., the size of the measured ceramic sinter is too small and does not occupy the entire surface of the measuring holder. Additionally, the following crystallographic structural parameters for the obtained ceramic materials were determined using the Rietveld method: lattice parameter and unit cell volume. It can be seen (Table 3) that lattice distortions and significant deviations from the pattern parameters appeared. The obtained XRD results show that the HPLT sintering technique allows for obtaining materials with high phase purity and double perovskite structure because for comparison, the studied materials in the form of polycrystalline powder had a much larger amount of BaWO_4_ phase. Introducing Eu^3+^ into the structure did not change the structural properties of the material.

Figure 5 shows EDS maps and Appendix A shows the percentage distribution of the elements on the sample surface of BMW:Eu-CST ceramics with different times of sintering. For the remaining materials, the results are almost identical, therefore it was decided to show the results only for one representative group of ceramic sinters—BMW:Eu-CST. The obtained ceramic materials are heterogeneous—an uneven distribution of elements in the structure is observed. There is significant precipitation of elements. In the place where Ba precipitation occurs, there are losses in W and inversely. Changing the parameter of the sintering process does not improve the homogeneity of the ceramic materials; one may be tempted to say that it worsens.

The Figure 6 shows the spectroscopic properties—the representative excitation and emission spectra of Eu^3+^ ions. The excitation spectrum (Figure 6 left) was monitored at 595 nm for ceramics sintered at 1 GPa and 500 °C. The spectrum contains two groups of bands: the first is the most intense and broad band, with a maximum at 300 nm associated with charge transfer (CT) transitions of O^2−^-Eu^3+^ and O^2−^-W^6+^, and the second is the narrower and less intense lines located at 370–400 nm associated with f-f transitions of the Eu^3+^ ions [19]. The results show that excitation by the matrix and CT are the most efficient way to obtain the emission of Eu^3+^ ions. The 300 nm line was used for excitation in the next experiments. Under 300 nm excitation, the obtained ceramics exhibit efficient red emission (Figure 6, right). The most efficient is the band centered at 595.5 nm assigned with magnetic dipole transition of Eu^3+ 5^D_0_ → ^7^F_1_. The intensity of this transition does not depend on the local symmetry [16]. The translucent properties of ceramics led to scattering of irradiation and homogeneous emission from the ceramic surface (Appendix A in Appendix A).

The influence of sintering temperature on the luminescence properties of ceramics (BMW:Eu-CT) is shown in Figure 7a. It was observed that the intensity of Eu^3+^ decreases with increasing annealing temperature. This is related to the sintering method of the ceramics and the diffusion of carbon deep into the ceramics as the sintering temperature increases, leading to quenching of the luminescence, by increasing the rate of the non-radiative process, as evidenced by reduced decay time (Figure 7d). The presence of carbon was confirmed by SEM-EDS analysis (see Appendix A in Appendix A). The decay times were shortened from 3.53 ms to 2.84 ms by increasing the annealing temperature from 800 to 1350 °C.

It is worth noting that the full width at half maximum (FWHM) increases with the temperature from 2.64 nm for the ceramic sintered at 500 °C to 3.02 for the ceramic sintered at 1350 °C (see Appendix A). This indicates a reduction in grain size associated with the decomposition of particle envelopes from the crystalline to the amorphous state, as described in [20]. The effect of narrowing of FWHM of Stark components of Eu^3+^ emission is observed also in the case of BMW:Eu-CP series (see Appendix A). The width of the emission band increases with applied pressure, but the effect is saturated when the pressure is higher than 4 GPa. However, in the case of BMW:Eu-CST, the sintering time does not affect the luminescence properties of Eu^3+^ ions.

The increasing pressure during sintering reduces the luminescence intensity (see Figure 7b) and does not significantly affect the luminescence decay dynamic (see Figure 7e), as the decay times are equal to 4.04 ms and 3.53 ms for samples sintered at 1 and 8 GPa, respectively. Such high values of decay time were also observed for BMW: Eu^3+^ powders [18]. The increase in sintering time from 1 to 3 min enhances the luminescence output (see Figure 7c), but when the ceramic is sintered for 5 min, the luminescence is quenched, most likely by the same processes as in the CT series (see Figure 7f), as the decay time is reduced from 2.66 ms to 1.52 ms by the presence of non-radiative processes.

In contrast to the excitation spectrum, the absorption spectrum measured in reflectance mode showed that only bands associated with CT transitions of O^2^-Eu^3+^ and O^2-^-W^6+^ were observed (Figure 7). According to previous reports on this material [19], abovementioned CT transitions are located at 300 nm and 320 nm, respectively. The intensity of the former is reduced in comparison with the powder material, indicating that the excitation by the energy transfer from the host is reduced by the sintering process. The f-f transitions have significantly lower intensity than the charge transfer transitions due to the high symmetry of the Eu^3+^ site.

## 4. Conclusions

Three series of ceramics prepared by the HPLT technique were studied to determine the effect of pressure (CP), sintering temperature (CT), and sintering time (CTS) on the structure and luminescence of Eu^3+^-doped BMW.

The sintering process conserves the high symmetry structure of double perovskite and the characteristic emission spectrum of BMW: Eu^3+^. The samples can be efficiently ex-cited at UV in the range of 200–330 nm through the CT and host absorption. The optimal sintering parameters for strong Eu^3+^ luminescence are T = 500 °C, *p* = 8 GPa, and t = 1 min. The increase in temperature results in increased quenching of the luminescence due to the diffusion of carbon deep into the ceramics. The increase in pressure leads to the creation of an amorphous phase, which enhances the rate of non-radiative transitions and also leads to the quenching of luminescence.

The obtained ceramics are non-homogeneous and the precipitation of elements in-creases with increasing pressure, temperature, and sintering time. To obtain homogeneous BMW ceramics other sintering methods are advised.

Due to the translucent properties of ceramics, the investigated matrix appears to be a good medium for new light sources pumped with semiconductor diodes.

## Figures and Tables

**Figure 1 materials-15-07410-f001:**
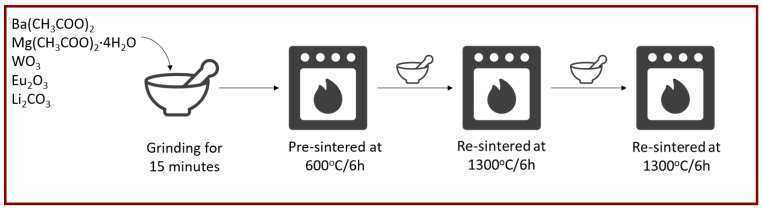
Solid-state synthesis scheme for obtaining polycrystalline BMW:Eu powders.

**Figure 2 materials-15-07410-f002:**
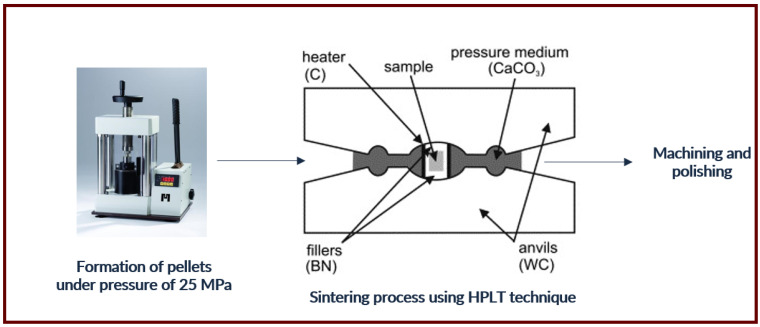
Schematic presentation of the toroidal high-pressure cell.

**Figure 3 materials-15-07410-f003:**
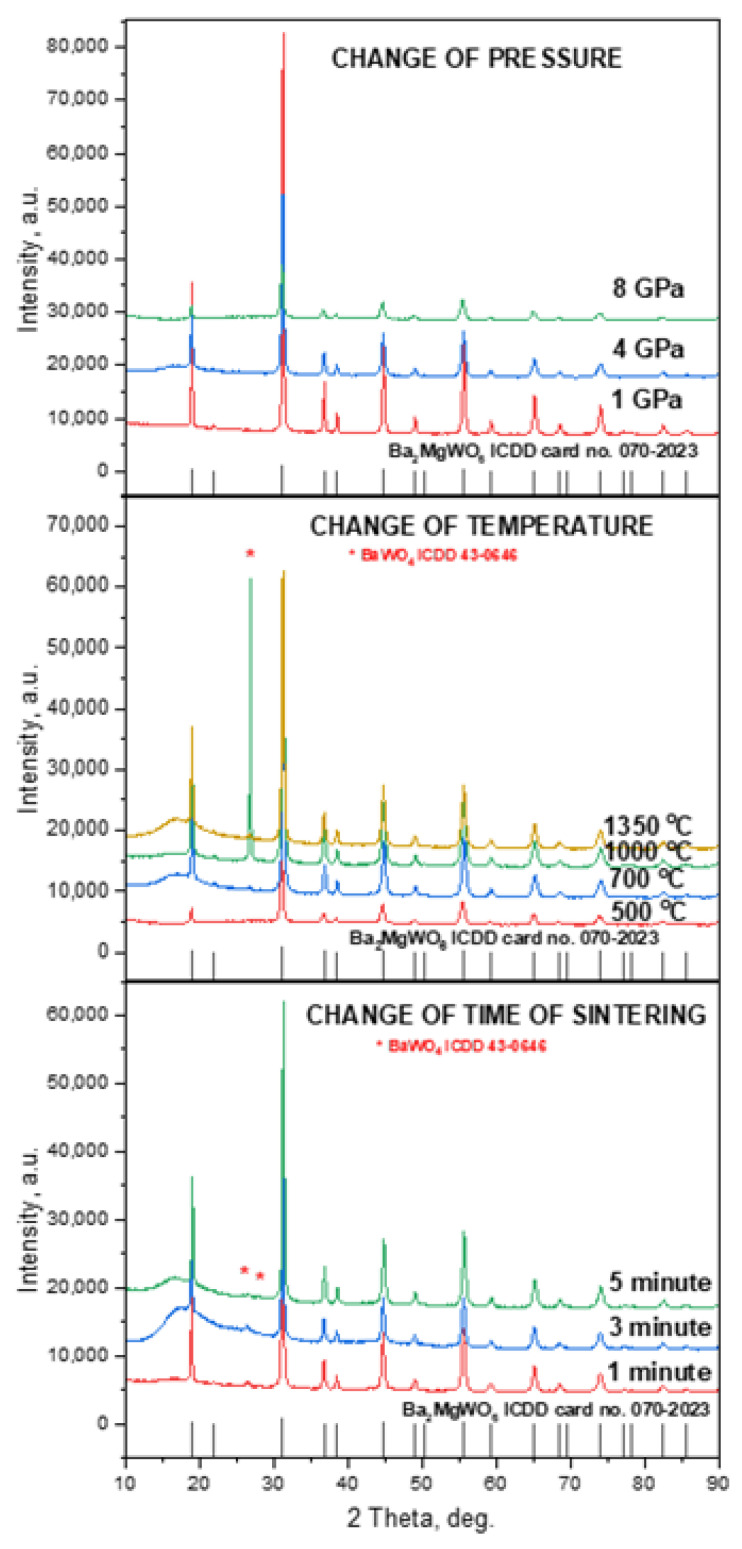
XRD patterns for BMW:Eu-CP, BMW:Eu-CT, and BMW:Eu-CTS, from up to down, respectively.

**Figure 4 materials-15-07410-f004:**
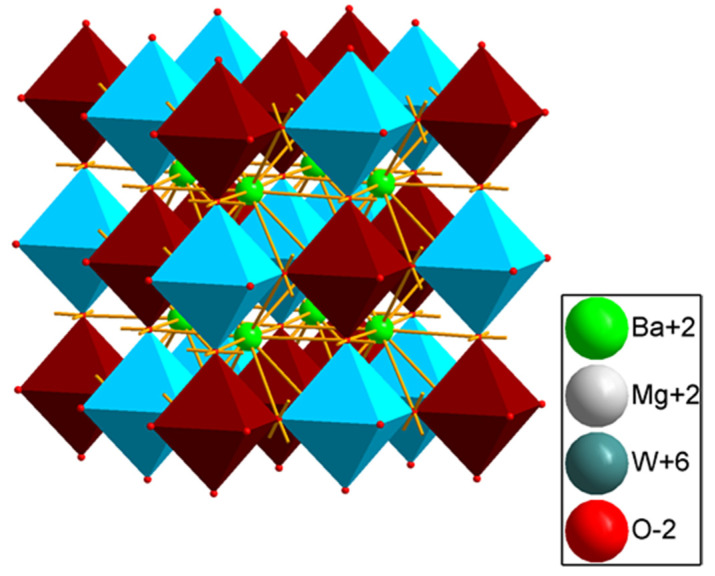
Crystal structure of ordered double perovskite Ba_2_MgWO_6_.

**Figure 5 materials-15-07410-f005:**
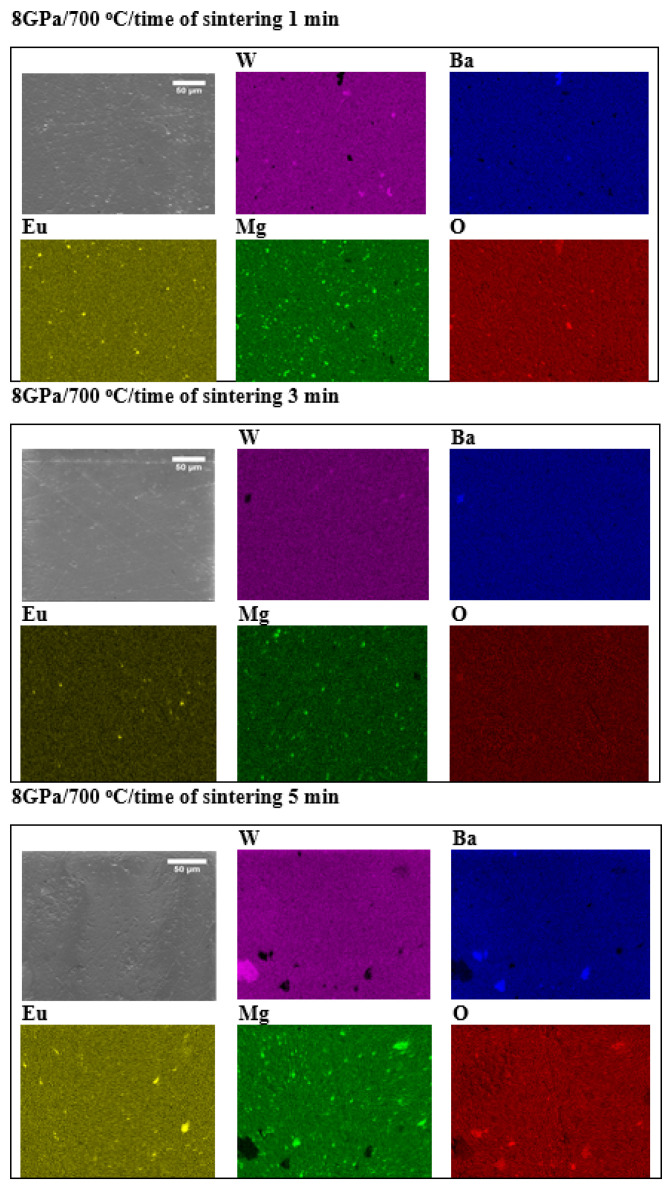
EDS maps of the BMW:Eu-CST ceramic materials (scale 50 µm).

**Figure 6 materials-15-07410-f006:**
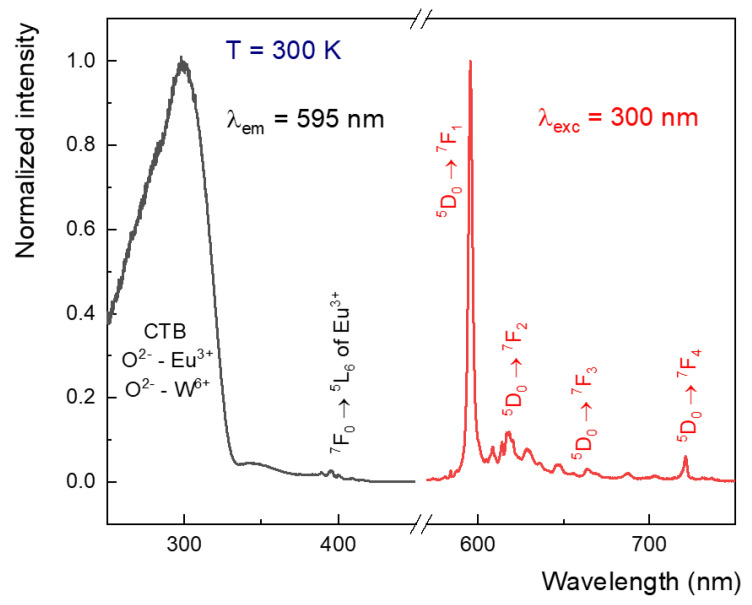
The excitation (black) and emission (red) spectrum of BMW:Eu ceramic materials sintered at 1 GPa and 500 °C.

**Figure 7 materials-15-07410-f007:**
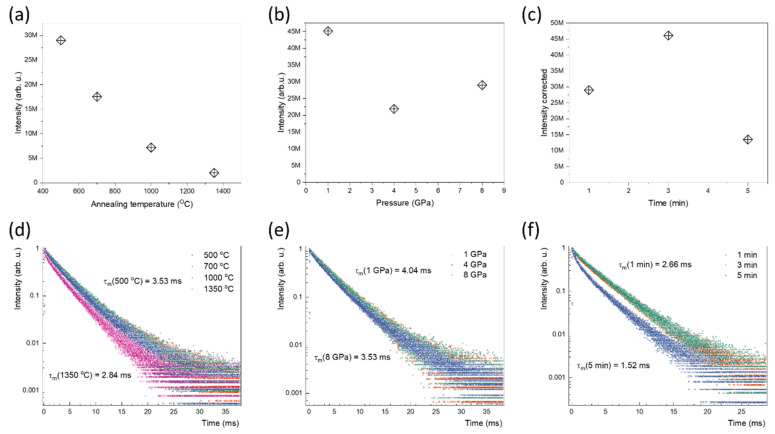
The luminescence intensity of BMW:Eu-CT (**a**), BMW:Eu-CP (**b**), and BMW:Eu-CTS (**c**) ceramic material series and the luminescence decay curves of BMW:Eu-CT (**d**), BMW:Eu-CP (**e**), and BMW:Eu-CTS (**f**).

**Table 1 materials-15-07410-t001:** Ionic radius values of individual ions in the host.

Ion in the Host	Coordination Number	
	6	8
Ba^2+^	-	156.0 pm
Mg^2+^	86.0 pm	-
W^6+^	74.0 pm	-
Eu^3+^	108.7 pm	120.6 pm

**Table 2 materials-15-07410-t002:** Parameters for the optimization of the sintering process.

Parameters	Changing Parameters		
	Pressure (CP)	Temperature (CT)	Time of Sintering (CTS)
*p*, (GPa)	1; 4; 8	8	8
T, (°C)	500	500; 700; 1000; 1350	700
t, (min)	1	1	1; 3; 5
r_T_, (°C/min)	600	600	600

**Table 3 materials-15-07410-t003:** Comparison of the crystallographic structure parameters of ceramic materials with the pattern.

Structural Parameters		Lattice Parameter a, (Å)	Unit Cell Volume, (Å^3^)
BMW Pattern		8.0901	529.49
CT	500 °C	8.12(1)	534.65
	700 °C	8.099(9)	531.19
	1000 °C	8.105(9)	532.51
	1350 °C	8.10(1)	532.17
CP	1 GPa	8.101(3)	531.67
	4 GPa	8.08(1)	528.46
	8 GPa	8.12(1)	534.65
CTS	1 min	8.09(1)	529.82
	3 min	8.09(6)	529.41
	5 min	8.106(6)	532.65

## Data Availability

Not applicable.

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
