# Peer review of "Influence of Sintering Parameters on Spectroscopic Properties of BMW: Eu3+ Ceramic Materials Prepared by HPLT Technique"

_materials, 2022, doi:10.3390/ma15217410_

Round 1

Reviewer 1 Report

The authors have conducted a comprehensive study about the effects of HPLT's pressure, temperature, and sintering time on phase evolution, elemental distribution, and spectroscopic performances of BMW:Eu3+ ceramic systems. The different processing conditions were discussed. The purpose and necessity of this study are fully defined. And the results related XRD, SEM/EDS and spectroscopic properties were deeply discussed which can support the conclusion. This paper is in good shape and can be accepted. But just a minor questions need to be answered:

1) Please check the type error. For example, P2L45: Chen ar.al [6] or Chen et al [6]; P4L137-139: please check the unit; Fig.5: please remove the red underline.

2) For Fig.5, where is the scale bar? Or where is the SEM images with scale bar (without EDS). Again, scale bar should be shown on surface morphology images so that readers can have a better understanding.

Author Response

Authors of the manuscript Ref. No.: materials-1951621 entitled: “Influence of sintering parameters on spectroscopic properties of BMW:Eu3+ ceramic materials prepared by HPLT technique” would like to thanks the Reviewers for the preparation of the review and their comments. These comments are valuable and very helpful for revising and improving our paper. We have studied reviewers’ comments carefully and tried our best to revise our manuscript according to the comments. All changes and new measurements developed in the manuscript are in red.

REVIEWER 1

The authors have conducted a comprehensive study about the effects of HPLT's pressure, temperature, and sintering time on phase evolution, elemental distribution, and spectroscopic performances of BMW:Eu3+ ceramic systems. The different processing conditions were discussed. The purpose and necessity of this study are fully defined. And the results related XRD, SEM/EDS and spectroscopic properties were deeply discussed which can support the conclusion. This paper is in good shape and can be accepted. But just a minor questions need to be answered:

ANSWER:

Thank you very much for reviewing our manuscript. We have addressed all of the listed points below. 

1) Please check the type error. For example, P2L45: Chen ar.al [6] or Chen et al [6]; P4L137-139: please check the unit; Fig.5: please remove the red underline.

ANSWER:

Thank you for the comments. We have included a new Figure 5 in the manuscript, corrected according to the reviewer's suggestions, as well as the aforementioned errors were corrected.

2) For Fig.5, where is the scale bar? Or where is the SEM images with scale bar (without EDS). Again, scale bar should be shown on surface morphology images so that readers can have a better understanding.

ANSWER:

Thank you for the comments. I agree that EDS maps should include scale. Therefore, they have been supplemented with SEM images that include the scale (Figure 5 below). The scale for SEM images and EDS maps are the same.

Figure 5. EDS maps of the BMW:Eu-CST ceramic materials (scale 50 µm).

I also agree with another suggestion of the reviewer that the work should be supplemented with SEM images in order to better define the morphological properties of the ceramic material. Below I present the SEM images made: SEM images of the BMW:Eu-CST ceramic materials. I did not include these images in the manuscript, because, first of all, we do not have a specialized electron microscope for imaging ceramics and therefore these images are not of the highest quality/resolution. Additionally, the obtained ceramic sinters were polished by hand, hence defects in the form of scratches and grooves may appear and are visible on the surface. Until now, samples have been polished in a foreign center, which has a specialized, automatic apparatus for polishing ceramic materials. Nevertheless, due to the present socio-political situation in the world, our contact was limited.

Summarizing the obtained SEM images (figure below), as a result of HPLT sintering, the materials became denser, no inter-grain boundaries are visible, and single pores are visible on the surface. The time of sintering has no effect on the morphological changes of the obtained ceramic sinters.

SEM images of the BMW:Eu-CST ceramic materials: left to right 1, 3 and 5 minutes, respectively.

Reviewer 2 Report

Miniajluk-Gawel et al. synthesized Eu3+ doped Ba2MgWO6 (BMW) double-perovskite ceramics and investigated spectroscopic properties. The HPLT technique with different sintering pressure, temperature, and time is utilized to obtain good luminescence performance of the BMW ceramics. They found that 500ºC, 1 GPa, and 3 min. were optimal condition for the HPLT sintering, which minimized the formation of the amorphous phase and the diffusion of carbon species into the ceramics.

1)   The Authors investigated the BMW ceramics sintered by the HPLT technique from the structural and spectroscopic points of view. The obtained BMW ceramics showed similar properties revealed from XRD and EDS analyses; the double-perovskite phase was formed, and the composition was not entirely homogeneous. In contrast, the Authors suggested that the spectroscopic properties depended on the HPLT sintering conditions. Although the Authors suggestion seemed correct to see Fig. 7, the readers would suspect that the HPLT sintering did not affect better spectroscopic properties. As shown in Fig. 7(a), the annealing temperature of 500ºC was optimal, but the luminescence intensity monotonically decreased with increasing annealing temperature. Such a result suggests that no annealing (no HPLT sintering) is the best for the BMW ceramics. I strongly recommend that the Authors show improvement of structural and luminescence properties by using the HPLT sintering at first and then optimizing the sintering conditions. Since an additional process takes cost, the Authors should demonstrate the improvement by applying the HPLT sintering. 

2)   In Line 95, please indicate the Li content in the sample composition like Ba1.99Li0.01Mg0.99Eu0.01O6.

3)   In Table 1, I do not know why the coordination number of 8 is used to discuss the ionic radii. The coordination number is 12 at the A site in a perovskite structure.

4)   Fig. 1  Fig. 4 (Line 162)

5)   The element distribution of the samples was discussed from the EDS mappings (Fig. 5). The sample is not homogeneous, and the homogeneity is not improved. It is natural because a few minutes of the sintering do not promote element migration. How about the homogeneity of the as-grown (not HPLT sintering) samples? 

6)   The comparison of the luminescence intensity (Figs. 7(a)–(c)) is strange to me. Is it possible to compare the intensity among samples? The plots of 500ºC in (a) and 8 GPa in (b) should be identical because of the identical sintering condition. Meanwhile, the intensity is greatly different. Although the axis is an arbitrary unit, normalization is essential to compare the intensity among the sample. Otherwise, the plot is meaningless. How did the Authors normalize the intensity in Figs. 7(a) and 7(b)?

Author Response

Authors of the manuscript Ref. No.: materials-1951621 entitled: “Influence of sintering parameters on spectroscopic properties of BMW:Eu3+ ceramic materials prepared by HPLT technique” would like to thanks the Reviewers for the preparation of the review and their comments. These comments are valuable and very helpful for revising and improving our paper. We have studied reviewers’ comments carefully and tried our best to revise our manuscript according to the comments. All changes and new measurements developed in the manuscript are in red.

REVIEWER 2

Miniajluk-Gaweł et al. synthesized Eu3+ doped Ba2MgWO6 (BMW) double-perovskite ceramics and investigated spectroscopic properties. The HPLT technique with different sintering pressure, temperature, and time is utilized to obtain good luminescence performance of the BMW ceramics. They found that 500ºC, 1 GPa, and 3 min. were optimal condition for the HPLT sintering, which minimized the formation of the amorphous phase and the diffusion of carbon species into the ceramics.

ANSWER:

Thank you very much for reviewing our manuscript. We have addressed all of the listed points below. 

1)   The Authors investigated the BMW ceramics sintered by the HPLT technique from the structural and spectroscopic points of view. The obtained BMW ceramics showed similar properties revealed from XRD and EDS analyses; the double-perovskite phase was formed, and the composition was not entirely homogeneous. In contrast, the Authors suggested that the spectroscopic properties depended on the HPLT sintering conditions. Although the Authors suggestion seemed correct to see Fig. 7, the readers would suspect that the HPLT sintering did not affect better spectroscopic properties. As shown in Fig. 7(a), the annealing temperature of 500ºC was optimal, but the luminescence intensity monotonically decreased with increasing annealing temperature. Such a result suggests that no annealing (no HPLT sintering) is the best for the BMW ceramics. I strongly recommend that the Authors show improvement of structural and luminescence properties by using the HPLT sintering at first and then optimizing the sintering conditions. Since an additional process takes cost, the Authors should demonstrate the improvement by applying the HPLT sintering. 

2)   In Line 95, please indicate the Li content in the sample composition like Ba1.99Li0.01Mg0.99Eu0.01O6.

ANSWER:

Thank you for the comments. In the experimental part it is described that substitution of a divalent ion with a third oxidation state lanthanide ion can cause defects in the matrix, which occur to compensate the created charges locally in the host. For this reason, charge compensation was applied through the addition of lithium in an amount of 50% based on the amount of dopant ions. The calculations were made by substituting lithium in position of magnesium, the same like the dopant ion. Nevertheless, the main assumption behind the use of lithium was charge compensation, and it really doesn't matter where it built into the structure and therefore it was not recorded in the formula of the structure obtained. Nevertheless, at the request of the reviewer, a summary formula of the obtained compound was given in the paper, taking into account the lithium content: Ba2Mg0.985Eu0.01Li0.005WO6.

3)   In Table 1, I do not know why the coordination number of 8 is used to discuss the ionic radii. The coordination number is 12 at the A site in a perovskite structure.

ANSWER:

Thank you for the comments. Of course, I agree that cation A is twelve times coordinated. Nevertheless, when we determining the position of the Eu3+ ion substitution in the structure (Ba2+, Mg2+, or W6+), we compare the distances between ionic radius values with the same coordination. The maximum coordination of Eu3+ ions is 8, therefore we took the same coordination number for Ba2+ ions. For example, Mg2+ ions are coordinated 6 times, therefore it was compared with Eu3+ ions with a coordination number of 6.

4)   Fig. 1 à Fig. 4 (Line 162)

ANSWER:

Thank you for the comments. The number has been changed.

5)   The element distribution of the samples was discussed from the EDS mappings (Fig. 5). The sample is not homogeneous, and the homogeneity is not improved. It is natural because a few minutes of the sintering do not promote element migration. How about the homogeneity of the as-grown (not HPLT sintering) samples? 

ANSWER:

Thank you for the comments. EDS mapping was not performed for the BMW:Eu polycrystalline powders from which the ceramic materials were sintered. However, in the previous work [Miniajluk-Gaweł, N.; Bondzior, B.; Lemański, K.; Vu, T. H. Q.; Stefańska, D.; Boulesteix, R.; Dereń, P. J. Effect of ceramics formation on the emission of Eu3+ and Nd3+ ions in double perovskite. Materials 2021, 14(20), 5996.] we presented a SEM image (presented below) of this material. The grains tend to agglomerate, but individual grains also appear. The grains have a relatively large size in the range 490 nm - 2 µm. SEM image analysis shows that solid state synthesis produces isotropic, slightly agglomerated and submicrometric Ba2MgWO6 double perovskite powders, and therefore these materials should exhibit good sinterability.

SEM images of the polycrystalline powder BMW-Eu.

6)   The comparison of the luminescence intensity (Figs. 7(a)–(c)) is strange to me. Is it possible to compare the intensity among samples? The plots of 500ºC in (a) and 8 GPa in (b) should be identical because of the identical sintering condition. Meanwhile, the intensity is greatly different. Although the axis is an arbitrary unit, normalization is essential to compare the intensity among the sample. Otherwise, the plot is meaningless. How did the Authors normalize the intensity in Figs. 7(a) and 7(b)?

ANSWER: The data have been revised and corrected. The Figure and the corresponding text have been changed.

Round 2

Reviewer 2 Report

I (Reviewer 2) was disappointed to see the Authors response letter and revised manuscript.

They did not answer my comment (1). In this comment, I strongly recommend the Authors' revisions to indicate any improvement of structural and luminescence properties by using the HPLT sintering and then optimizing the sintering conditions. Nevertheless, the Authors ignored it. 

Furthermore, the Authors responded to my comment (6), "The Figure and corresponding text have been changed." However, I cannot find the Authors' revisions in Figs. 7(a) and 7(b). In addition, they did not reply to my comment of "How did the Authors normalize the intensity in Figs. 7(a) and 7(b)?". 

These dishonest responses cannot be acceptable in a scientific manner. The Authors should answer all questions raised by the referees as much as possible. Owing to the Authors' insufficient reply and revision, I cannot recommend the manuscript for publication in Materials.

Author Response

I (Reviewer 2) was disappointed to see the Authors response letter and revised manuscript.

ANSWER:

Thank you very much for reviewing our manuscript. We sincerely apologize for our oversight. Below I have listed the answers to the reviewer's questions.

They did not answer my comment (1). In this comment, I strongly recommend the Authors' revisions to indicate any improvement of structural and luminescence properties by using the HPLT sintering and then optimizing the sintering conditions. Nevertheless, the Authors ignored it. 

 ANSWER:

Thank you for the comments. Of course, I agree with the opinion of the reviewer that both the synthesis of preparation and the sintering conditions should be optimized, which was also done. Our previous work [1] describes quite extensively the method of obtaining polycrystalline powders by the solid state method, which were then used for sintering ceramic materials. Of course, the solid state method was previously optimized by selecting the appropriate precursors, excess Mg amount, temperature and time of annealing. It was performed at the stage of laboratory work, on the basis of which the most optimal conditions for the synthesis of polycrystalline powders by the solid state method were selected. The HPLT sintering conditions have also been optimized, as is the case of the work, by selecting the optimal pressure, temperature, time and sintering rate.

By using HPLT sintering, an improvement in the structural and spectroscopic properties of the obtained ceramic materials is observed. Below I will try to present the differences in properties between polycrystalline powders and obtained ceramic materials, so as to prove the correctness of the HPLT sintering technique used.

The obtained XRD results (below, on the left XRD patterns of the polycrystalline powders and on the right XRD patterns of ceramic materials) show that the HPLT sintering technique allows for obtaining materials with high phase purity and double perovskite structure because for comparison, the studied materials in the form of polycrystalline powder had much larger amount of BaWO4 phase.

 Figure 2. XRD patterns of the polycrystalline powders: BMW, BMW-Eu and BMW-Nd [1].

Figure 3. XRD patterns for BMW:Eu-CP, BMW:Eu-CT and BMW:Eu-CTS, from up to down respectively.

Below I present the SEM images made for ceramic materials: SEM images of the BMW:Eu-CST ceramic materials. I did not include these images in the manuscript, because, first of all, we do not have a specialized electron microscope for imaging ceramics and therefore these images are not of the highest quality/resolution. Additionally, the obtained ceramic sinters were polished by hand, hence defects in the form of scratches and grooves may appear and are visible on the surface. Until now, samples have been polished in a foreign center, which has a specialized, automatic apparatus for polishing ceramic materials. Nevertheless, due to the present socio-political situation in the world, our contact was limited.

Summarizing, the grains of polycrystalline powders tend to agglomerate, but individual grains also appear. The grains have a relatively large size in the range 490 nm - 2 µm. SEM image analysis shows that solid state synthesis produces isotropic, slightly agglomerated and submicrometric Ba2MgWO6 double perovskite powders, and therefore these materials should exhibit good sinterability. The obtained SEM images of ceramic materials suggests, that the HPLT method is suitable for the sintering process. The materials became denser, no inter-grain boundaries are visible, and single pores are visible on the surface. The time of sintering has no effect on the morphological changes of the obtained ceramic sinters.

SEM images of the BMW:Eu-CST ceramic materials: left to right 1, 3 and 5 minutes, respectively.

SEM images of the polycrystalline powder BMW-Eu.

Furthermore, the Authors responded to my comment (6), "The Figure and corresponding text have been changed." However, I cannot find the Authors' revisions in Figs. 7(a) and 7(b). In addition, they did not reply to my comment of "How did the Authors normalize the intensity in Figs. 7(a) and 7(b)?". 

ANSWER:

The authors agree with the reviewer that the comparison of intensities between different samples is a debatable thing, especially in polycrystalline samples, where the scattering of radiation affects the obtained result.

In the case of the results presented here, measurements were made from the polished surface of a high-density ceramic, and the size of the excitation spot was smaller than the size of the sample, to minimize the error resulting from the comparison of emission intensities.

The Figure 7 was mistakenly left unchanged. In the current version the correct Figure was included, in which the intensity was calculated based on not normalized spectra and the values of intensity are consistent among three Figures.

Sintering of ceramics, does not significantly affect the emission spectroscopic properties, but broadening the half-width of the emission line has been observed (Fig. S4 and S5). The studied materials, due to their cubic structure, are good candidates for obtaining transparent ceramics. Due to the difficulty of obtaining a polycrystalline precursor with high purity, using the HPLT sintering method, translucent ceramics are obtained. Thus, this method makes it possible, in a fast and cheap way, to obtain a material with high density, high mechanical strength, while preserving the spectroscopic properties of the precursor. As shown by the sintering optimization process, conditions of 500 °C, 8GPa and 1 minute are optimal for this material.

As shown in Fig. S8, this material, due to its translucent properties, scatters radiation well, and exhibits whole-volume emission, giving great potential for use as a matrix for LED-pumped light sources.
